# Investigation of Alkali-Silica Reactivity in Sustainable Ultrahigh Performance Concrete

**Safeer Abbas [1], Wasim Abbass [1], Moncef L. Nehdi [2],\*, Ali Ahmed [1] and Muhammad Yousaf [1]**

[1] Civil Engineering Department, University of Engineering and Technology Lahore, Lahore 54890, Pakistan; safeer.abbas@uet.edu.pk (S.A.); wabbass@uet.edu.pk (W.A.); ali@uet.edu.pk (A.A.); yousaf.engr786@gmail.com (M.Y.)

[2] Department of Civil and Environmental Engineering, Western University, London, ON N6A 5B9, Canada

\* Correspondence: mnehdi@uwo.ca; Tel.: +1-519-6612111 (ext. 88308)

**Abstract:** Considering its superior engineering properties, ultrahigh performance concrete (UHPC) has emerged as a strong contender to replace normal strength concrete (NSC) in diverse construction applications. While the mechanical properties of UHPC have been thoroughly explored, there is still dearth of studies that quantify the durability of UHPC, especially for sustainable mixtures made with local materials. Therefore, this research aims at investigating the alkali-silica reactivity (ASR) potential in sustainable UHPC in comparison with that of NSC. Sustainable UHPC mixtures were prepared using waste untreated coal ash (CA), raw slag (RS), and locally produced steel fibers. UHPC and benchmark NSC specimens were cast for assessing the compressive strength, flexural strength, and ASR expansion. Specimens were exposed to two curing regimes: accelerated ASR conditions (as per ASTM C1260) and normal water curing. UHPC specimens incorporating RS achieved higher compressive and flexural strengths in comparison with that of identical UHPC specimens made with CA. ASR expansion of control NSC specimens exceeded the ASTM C1260 limits (>0.20% at 28 days). Conversely, experimental results demonstrate that UHPC specimens incurred much less ASR expansion, well below the ASTM C1260 limits. Moreover, UHPC specimens incorporating steel fibers exhibited lower expansion compared to that of companion UHPC specimens without fibers. It was also observed that the mechanical properties of NSC specimens suffered more drastic degradation under accelerated ASR exposure compared to UHPC specimens. Interestingly, UHPC specimens exposed to accelerated ASR conditions attained higher mechanical properties compared to that of reference identical specimens cured in normal water. Therefore, it can be concluded that ASR exposure had insignificant effect on sustainable UHPC incorporating CA and RS, especially for specimens incorporating fibers. Results indicate that UHPC is a robust competitor to NSC for the construction of mega-scale projects where exposure to ASR conducive conditions prevails.

**Keywords:** ultrahigh performance concrete; normal strength concrete; alkali-silica reaction; expansion; strength degradation; coal ash; raw slag; steel fiber

## 1. Introduction

Recent developments and advances in chemical and mineral admixtures have led to the introduction of various types of ultra-durable cement based materials. A noteworthy innovation in concrete technology is the development of ultrahigh performance concrete (UHPC) incorporating various supplementary cementitious materials (SCMs) and fibers [1]. It can be defined as a cement-based composite having a minimum compressive strength of 150 MPa and superior tensile and flexural strengths, enhanced durability and toughness [2–4]. Typically, UHPC consists of a high dosage of Portland cement, silica fume, fine sand (normally quartz), chemical admixture (new generation superplasticizer), and fibers, and is produced at a very low water to binder ratio (i.e., 0.15–0.25) [4]. Microstructural improvement, enhanced homogeneity of the mixture, reduced porosity through improved particle packing

density, excellent hydration, and better toughness are the important principles to produce UHPC on large scale [4,5]. Unlike normal strength concrete (NSC), coarse aggregates are not normally included in UHPC mixtures. Consequently, eliminating microcracks present in coarse aggregates and the typically weak aggregate-cement paste interfacial transition zone help attaining ultrahigh strength properties [6]. The superior mechanical strength of UHPC makes it a material of choice for producing prefabricated structural members that are slenderer, occupying less space, and having reduced weight [6,7].

Moreover, the excellent durability characteristics of UHPC mixtures make it a preferred option for longer service life performance, especially in severe exposure conditions [5,8]. On the other hand, the self-healing characteristics owing to the abundantly available un-hydrated cement particles in UHPC can lead to the development of more sustainable and resilient infrastructure in comparison to that made of NSC [8,9].

Extensive research has been carried out on the design of UHPC mixtures [10–12], its rheological properties [13,14], fiber matrix interaction properties [15–17], mechanical properties, along with dimensional stability [18–21] and microstructural characterization [22,23]. Similarly, various governmental agencies have supported large-scale research projects for the development of UHPC guidelines, such as the French Association of Civil Engineers, Japan Society of Civil Engineers, and the American Concrete Institute [4,24,25]. Further, various mega-structures have already been constructed using this innovative UHPC material, including various buildings, bridges, utilities for the oil and gas industry, hydraulic structures, and in the repair of existing structures [26,27].

However, research on the alkali-silica reaction (ASR) in UHPC is still limited. ASR is one of the major durability concerns leading to premature degradation and costly maintenance and repair in concrete structures. It consists of a reaction between alkalis in the concrete and reactive silica in aggregates in the presence of moisture, leading to the formation of ASR gel [28]. This gel absorbs the surrounding water and expands in volume, which exerts tensile stresses on the surrounding concrete, and causes cracking and spalling [29–31]. The major factors affecting ASR include the type and amount of reactive silica (amorphous, cryptocrystalline or crystalline silica) in the aggregate, concentration of alkalis in the pore solution, and the availability of sufficient water [32].

Various structures have been severely damaged by ASR around the globe. The first ASR damaged structure reported was the Parker Dam in Arizona in 1941 [33]. Later, various structures were diagnosed to be affected by ASR, including for instance the turbine foundation of the Ikata power plant in Japan [34], Bibb Graves Bridge in Albama, USA [35], Seabrook Nuclear Power Plant in New Hampshire, USA [36], Vosnasvej Bridge in Denmark [37], Robert-Bourassa/Charest overpass in Quebec, Canada [38], and the Warsak Dam in Pakistan [39,40]. The deleterious effects of ASR in concrete structures can be minimized by controlling the alkali content, minimizing reactive silica, reducing the exposure to water, and addition of alkali-silica expansion inhibitor admixtures. Moreover, the partial replacement of cement with an adequate amount and type of supplementary cementitious materials can be highly effective in controlling ASR expansion and the associated damage [41–43]. For instance, Shafaatian et al. (2013) [44] reported that the replacement of cement with supplementary cementitious materials (SCMs) limits the amount of alkalis in the paste, thus reducing the ASR expansion.

It can be argued that the durability of concrete structures is an important parameter that affects and dictates the overall service life of structures. Moreover, the life cycle cost of structures is a function of durability indicators. It was reported that the use of SCMs can reduce the overall porosity of the mixture, which inhibits the intrusion of moisture and other aggressive species inside the concrete, thus helping to mitigate the ASR associated durability aspects [45,46]. Another method to improve the durability related issues of concrete is to apply coatings such as epoxy at the external concrete surfaces, which act as a barrier against the intrusion of harmful materials inside the concrete [47].

Limited studies have investigated the ASR potential of UHPC. For instance, Graybeal et al. (2006) [48] explored the ASR potential of UHPC specimens having $25 \times 25 \times 280$ mm in

size and exposed to a sodium hydroxide (NaOH) solution for 28 days at 80 °C, as per the ASTM C1260 guidelines. The expansion reported after 28 days was 0.012% and 0.004% for air and steam cured specimens, respectively, which is below the ASTM C1260 specified limits (<0.2%) [48]. It was also reported that upon incorporating various contents of quartz sand (0%, 50%, and 100% as aggregate replacement) in UHPC, the expansion of specimens was around 0.03% at 16 days [49]. Moser et al. (2008) [50] reported that the expansion of pre-damaged and undamaged UHPC specimens showed ASR expansion of 0.02% after 600 days [50].

The original contribution of this study consists of the production of sustainable UHPC incorporating locally available raw materials and using the normal curing regime adopted in industrial precast plants instead of high thermal curing. Moreover, there is dearth of studies on the ASR potential of UHPC incorporating untreated coal ash (CA) and raw slag (RS). In addition, the steel fibers used in the present study were cut from long recyclable steel wires, rather than the costly conventional steel fibers that are commercially available. Hence, this research program was planned to investigate the ASR expansion and its possible deleterious damage in this sustainable and locally produced UHPC incorporating untreated waste materials. Furthermore, the novelty of this study lies in evaluating the mechanical properties of the produced UHPC specimens exposed to ASR conducive condition (1N NaOH at 80 °C) as compared to that of identical control specimens cured in water. The comparison of the performance of UHPC with that of NSC under ASR conditions can provide insight into the potential application of sustainable UHPC in large-scale construction exposed to ASR conducive conditions. This study was envisioned to assist stakeholders and decision makers in appraising UHPC incorporating local waste materials for the design of sustainable, resilient and durable civil infrastructure under severe ASR exposure conditions. The outcome of this study will benefit researchers and other construction stakeholders for better understanding the improved durability performance of sustainable UHPC mixtures, especially in aggressive exposures. Moreover, the utilization of sustainable UHPC mixtures incorporating waste materials reduces the environmental overburden due to the deposition of byproducts in open landfills.

## 2. Materials and Methods

### 2.1. Materials and Specimen Preparation

Ordinary Portland cement was used. Untreated coal ash (CA) was acquired from a local thermal power plant. Raw slag (RS) was also procured from local steel industry. Fibers of 10 mm long were prepared by cutting long-recycled steel wire (Figure 1).

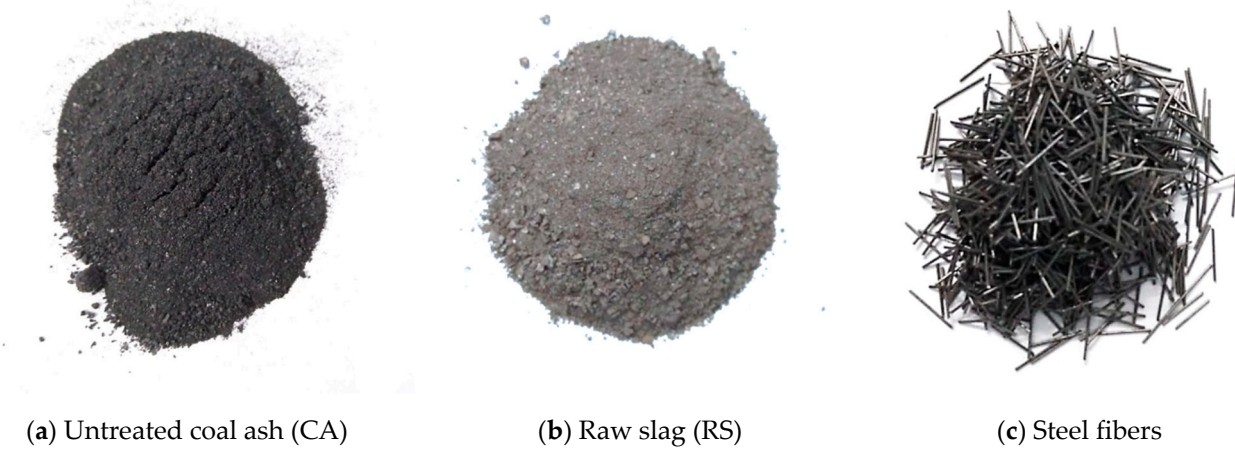

(**a**) Untreated coal ash (CA)　　　　(**b**) Raw slag (RS)　　　　(**c**) Steel fibers

**Figure 1.** Raw materials used.

Scanning electron microscopy (SEM) images of CA and RS are shown in Figure 2. It was observed that CA particles were porous in nature, while RS particles were denser and more compact. Table 1 provides chemical analysis of the used cement, CA and RS.

Both CA and RS showed loss on ignition (LOI) higher than 10%. Table 2 shows the physical properties of used cement, CA and RS. Silica sand was obtained from the local glass industry, having $SiO_2$ greater than 98%. Quartz powder having grain size of 450 to 500 microns was also used. Polycarboxylate based superplasticizer (SP) was added to control the workability of mixtures.

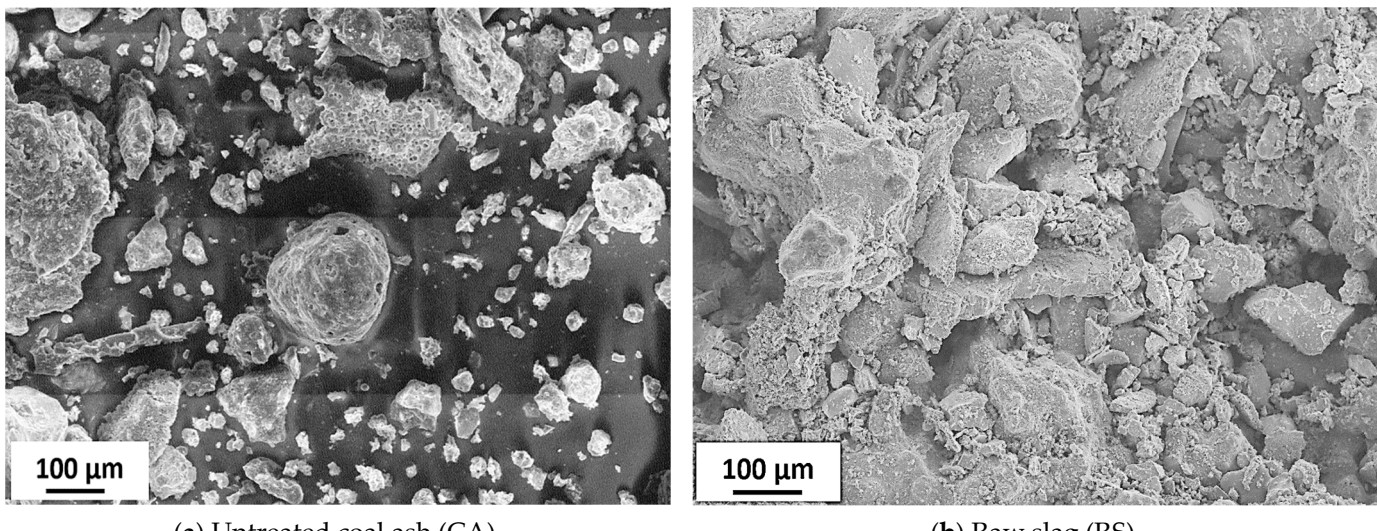

(**a**) Untreated coal ash (CA)        (**b**) Raw slag (RS)

**Figure 2.** Microstructure of CA and RS.

**Table 1.** Chemical properties of used raw materials.

| Materials | CaO | $Al_2O_3$ | $Fe_2O_3$ | $SiO_2$ | MgO | $SO_3$ | LOI |
|---|---|---|---|---|---|---|---|
| Cement (%) | 60.05 | 3.79 | 5.54 | 20.15 | 3.10 | 2.76 | 2.90 |
| CA (%) | 9.45 | 12.27 | 4.63 | 36.85 | 1.52 | 3.96 | 11.05 |
| RS (%) | 7.65 | 11.15 | 4.78 | 32.85 | 0.51 | 4.88 | 10.85 |

**Table 2.** Physical properties of used raw materials.

| Properties | Cement | CA | RS |
|---|---|---|---|
| Unit weight (kg/m$^3$) | 1542 | 360 | 1208 |
| Fineness (Passing 200 sieve) (%) | >95 | >95 | >95 |
| Blaine fineness (cm$^2$/g) | 3010 | 4344 | 4536 |
| Specific gravity | 3.10 | 2.32 | 2.27 |

Table 3 shows the mixture design of the tested UHPC mixtures. Initially, cement along with CA, RS, and sand were dry mixed for one minute in a high-speed rotating shear mixer. Before adding the dry ingredients of UHPC, the mixer bowl was rinsed with wet cloth. Half of the amount of the calculated water was progressively added in the mixture. The remaining water was mixed with the superplasticizer and added into the mixture, followed by continuous high-shear mixing for another three minutes. Steel fibers were added last in the mixture and mixing resumed till a homogenized mixture with well dispersed fibers and fiber clumping was achieved. Afterwards, specimens for compressive and flexural strengths and expansion measurements were prepared and consolidated on a vibrating table. Cube, prism, and mortar bar specimens were made with five replicates for each test at each desired test age. All specimens were covered with plastic sheets and placed in a moist curing chamber at 45 °C and relative humidity greater than 95%. After 24 h, specimens were taken out from their respective molds and cured in water at 45 °C for another 24 h. Thereafter, specimens were separated in two exposure conditions. The first

set of specimens (cube, prism, and mortar bars) were exposed to 1N NaOH solution at $80 \pm 5\,^{\circ}\text{C}$, while the remaining specimens were cured in water at $20\,^{\circ}\text{C} \pm 2$.

**Table 3.** Mixture design of tested UHPC.

| Materials | Mass/Cement Mass | | | |
|---|---|---|---|---|
| | **UHPC1** | **UHPC2** | **UHPFRC1** | **UHPFRC2** |
| Cement | 1.00 | 1.00 | 1.00 | 1.00 |
| Fine sand | 0.50 | 0.50 | 0.50 | 0.50 |
| Silica sand | 0.55 | 0.55 | 0.55 | 0.55 |
| Quartz powder | 0.28 | 0.28 | 0.28 | 0.28 |
| Untreated coal ash | 0.20 | - | 0.20 | - |
| Raw slag | - | 0.20 | - | 0.20 |
| Steel fibers | - | - | 0.20 | 0.20 |
| SP | 0.04 | 0.04 | 0.04 | 0.04 |

Cement = 780 kg/m$^3$; water to binder ratio = 0.28.

Furthermore, cube, prism, and mortar bar specimens of normal strength concrete (NSC) were also made for comparison with the ultrahigh performance concrete (UHPC). For preparing the NSC specimens, aggregates were procured from a local crush quarry. Specimens were prepared using a cement-to-aggregate ratio of 1 to 2.25 as per the ASTM C1260 guidelines [51]. Furthermore, the grading of the aggregate used for preparing the specimens was similar to that in ASTM C1260. The water-to-cement ratio was 0.475. These NSC specimens were also placed in the two previously mentioned curing regimes. Table 4 shows the number of specimens and their coefficient of variance (COV) for various conducted tests.

**Table 4.** Number of specimens and their coefficient of variance for various tests.

| Mixtures | Age (Days) | Number of Specimens | Compressive Strength | | Flexural Strength | | Expansion |
|---|---|---|---|---|---|---|---|
| | | | **Water** | **ASR** | **Water** | **ASR** | **ASR** |
| UHPC1 | 14 | 5 | 1.12 | 1.51 | 0.90 | 0.76 | 0.74 |
| | 28 | 5 | 1.76 | 1.42 | 1.14 | 1.41 | 0.61 |
| | 90 | 5 | 1.36 | 1.13 | 1.21 | 1.48 | 0.56 |
| UHPC2 | 14 | 5 | 1.21 | 1.39 | 0.78 | 0.91 | 0.69 |
| | 28 | 5 | 1.82 | 1.72 | 1.21 | 1.71 | 0.52 |
| | 90 | 5 | 1.41 | 1.83 | 1.34 | 1.47 | 0.47 |
| UHPFRC1 | 14 | 5 | 1.81 | 1.74 | 1.32 | 1.98 | 0.76 |
| | 28 | 5 | 1.92 | 1.81 | 1.45 | 1.68 | 0.62 |
| | 90 | 5 | 1.98 | 1.89 | 1.64 | 2.10 | 0.39 |
| UHPFRC2 | 14 | 5 | 1.91 | 1.92 | 1.76 | 2.05 | 0.69 |
| | 28 | 5 | 1.79 | 1.79 | 1.81 | 1.84 | 0.56 |
| | 90 | 5 | 1.87 | 1.82 | 1.91 | 1.95 | 0.41 |
| NSC | 14 | 5 | 1.92 | 1.95 | 2.15 | 2.23 | 0.92 |
| | 28 | 5 | 2.01 | 1.89 | 2.05 | 2.08 | 0.77 |
| | 90 | 5 | 1.98 | 2.02 | 2.24 | 2.19 | 0.83 |

*2.2. Test Procedures*

The flowability of fresh UHPC mixtures was measured using a flow table in accordance with ASTM C230 [52]. The compressive strength of UHPC mixtures was determined on cube specimens having $50 \times 50 \times 50$ mm in size, as per ASTM C109 [53]. The loading rate on specimens was 1.10 MPa/sec. The flexural strength of UHPC was measured on prism specimens having size $40 \times 40 \times 160$ mm in size, in accordance with ASTM C348 [54]. The loading rate for flexural strength testing was 2.70 kN/min. Mortar bars of $25 \times 25 \times 285$ mm in size were used to measure the expansion following ASTM C1260 provisions. At the desired age (3, 7, 14, 21, 28, 56, and 90 days), specimens were taken

out from the ASR exposure and water curing conditions and tested. For measuring each expansion reading, a digital length comparator was calibrated using a reference rod. The expansion measurements were conducted following the ASTM C490 procedure [55]. Each specimen was visually inspected prior to testing for investigating the presence of any surface distress or initiation of cracks. Furthermore, microstructural analysis was conducted on selected specimens using scanning electron microscope.

## 3. Results and Discussion

### 3.1. Flowability of UHPC

Table 5 shows the flow results of the tested UHPC mixtures. It was observed that the UHPC mixture incorporating CA attained lower values of flow compared to that of the UHPC mixture with RS. Relatively decreased flow of the UHPC mixture with CA was mainly attributed to its high loss on ignition content primarily consisting of unburnt carbon, which tends to compromise the effect of the superplasticizer. UHPC mixtures incorporating CA attained less flow also due to its porous microstructure, leading to increased water absorption during mixing [56]. Similar findings were reported in previous study [57].

**Table 5.** Flow of tested UHPC mixtures.

| Mixtures | UHPC1 | UHPC2 | UHPFRC1 | UHPFRC2 |
|---|---|---|---|---|
| Flow (mm) | 186 | 195 | 169 | 177 |

Furthermore, the results indicate that the addition of steel fibers reduced the flowability of UHPC mixtures, as expected. There was a reduction of around 10% in the flowability of UHPC mixtures incorporating steel fibers. These results agree with other previous findings [58,59]. The reduction in flow of UHPC mixtures incorporating steel fibers can be attributed to altering the skeleton structure of granular particles. This will cause increased friction and obstruction and hindrance of flow. Further, steel fibers act as a barrier to reduce the free-flowing velocity of fresh UHPC mixtures [59,60].

### 3.2. Compressive and Flexural Strength

Figure 3a depicts the compressive strength results of UHPC incorporating CA, RS, and steel fibers up to 90 days. The results show the average value obtained on five identical specimens with a coefficient of variance (COV) less than 2%. It was observed that the compressive strength of UHPC mixture incorporating fiber (UHPFRC2) was 141 and 150 MPa at 28 and 90 days, respectively. The compressive strength at 3, 7, 14, 21, 28, 56, and 90 days for the UHPC mixture incorporating CA (UHPC1) was 58, 81, 86, 95, 104, 109, and 114 MPa, respectively. This gain in compressive strength with time was due to continuous hydration of cement, as expected. The results revealed that the UHPC mixture incorporating CA (UHPC1) achieved 59% of the 28 days compressive strength at an early age of 3 days. This rate of gain in compressive strength is comparable with previous findings [58,61]. UHPC mixtures are well known to exhibit higher rate of gain in strength at early age [62]. Bahedh and Jaafar (2018) reported around 60% of 28 days compressive strength at 3 days for a mixture incorporating 20% of fly ash [61]. The UHPC mixture made with CA (UHPC1) achieved around 80% of the 28 days compressive strength at 7 days. Furthermore, the compressive was enhanced by 5% and 10% at 56 and 90 days, respectively, compared to that of at 28 days. This can be attributed to densification of the pore structure in the presence of CA particles at later age owing to pozzolanic reactions and the filing of micropores owing to formation of secondary calcium silicate hydrates (CSH) at later ages, leading to increased compressive strength [61,63]. Furthermore, it was observed that UHPC mixture made with RS (UHPC2) attained compressive strength values of 65, 90, 96, 110, 119, 123, and 126 MPa at 3, 7, 14, 21, 28, 56, and 90 days, respectively. The results show that UHPC2 achieved higher compressive strength compared to that of UHPC1. For example, UHPC2 attained 119 MPa compared to 104 MPa for UHPC1 at 28 days. The enhanced compressive strength of UHPC mixtures incorporating RS can be

attributed to higher fineness of RS particles in comparison with CA. SEM images of CA and RS confirm that RS had denser and more compact microstructure as compared to the porous and spongy structure of CA (Figure 2).

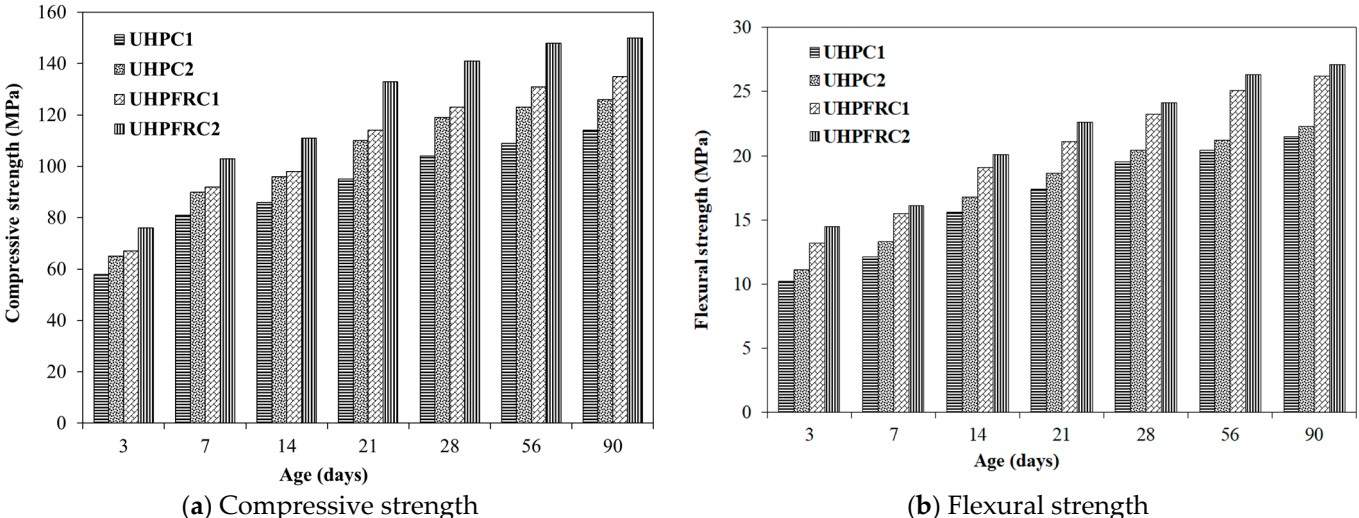

(**a**) Compressive strength        (**b**) Flexural strength

**Figure 3.** Compressive and flexural strength results for the various UHPC mixtures.

An increase in compressive strength was observed for mixtures incorporating steel fibers. For instance, the compressive strength of UHPFRC1 and UHPFRC2 was 67 MPa and 76 MPa at 3 days, respectively, corresponding to 16% increase compared to that of identical specimen without steel fibers. Similarly, the compressive strength of UHPFRC1 and UHPFRC2 was 92 MPa and 103 MPa at 7 days, respectively, similar to findings reported by others [62]. Such increase in compressive strength owing to steel fiber addition can be ascribed to enhanced confinement, leading to restrained lateral deformation [62]. Furthermore, a strong fiber–matrix bond was observed through SEM analysis (Figure 4), indicating the ability to reduce stress concentrations, and restraining internal material degradation, thus leading to improved strength.

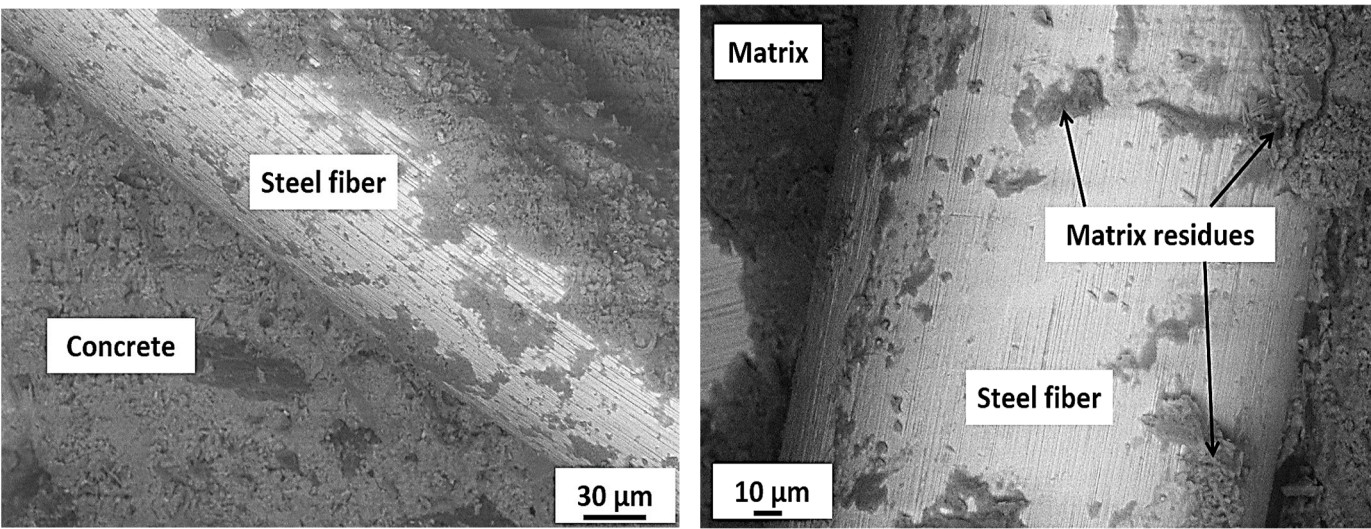

**Figure 4.** SEM images of UHPC specimen incorporating steel fibers.

Wu et al. (2019) [21] reported an increase in compressive strength of around 20% for UHPC mixtures incorporating steel fibers at 28 days compared to that of the mixtures without steel fibers [21]. This increase in compressive strength can be attributed to the fact

that steel fibers increase the rigidity of the matrix by restricting the initiation of micro-cracks prior to the peak load [64–66]. It was observed that UHPC specimens without steel fibers failed in an abrupt manner with sudden loud sound; however, specimens incorporating fibers remained intact without any concrete splitting or breaking.

Figure 3b also shows the flexural strength of UHPC mixtures incorporating CA and RS with and without fibers at different ages. Each result depicted in the figure is the average value of five specimens with COV of less than 3%. It can be observed that the flexural strength of UHPC mixtures increased with the curing age, as expected owing to continuous hydration of cement and pozzolanic reaction of CA and RS. For mixture UHPC1, the flexural strength at 7 and 28 days was 12.1 and 19.5 MPa, respectively. Similarly, the flexural strength of UHPC2 at 7 and 28 days was 13.3 and 20.4 MPa, respectively. The results indicate that UHPC mixtures incorporating CA and RS achieved around 65% of the 28 days flexural strength at 7 days. The UHPC mixture incorporating CA attained lower flexural strength in comparison with that of the UHPC mixture made with RS. Similar trend was observed earlier for the compressive strength results.

It was also observed that the addition of steel fibers increased the flexural strength for UHPC mixtures incorporating CA and RS. The flexural strength of UHPFRC1 at 7 and 28 days was 15 and 23 MPa, respectively. Similarly, UHPFRC2 achieved flexural strength at 7 and 28 days of 16 and 24 MPa, respectively. There was an increase in flexural strength of around 20% owing to addition of steel fibers in UHPC mixtures. Similar findings have been reported in previous studies [67,68]. The increase in flexural strength via addition of steel fibers can be ascribed to the strong bond of fibers to the cementitious matrix (Figure 4), leading to enhanced control of the initiation and propagation of microcracks as reported by others [56,62,67]. The presence of fibers in the matrix permits to bridge micro-cracks and restrain their growth before the initial peak load, ultimately contributing towards mechanical strength enhancement of UHPC [15]. Furthermore, Wu et al. (2016) [15] reported that increased development of CSH and strengthened interfacial transition zone around fibers enhanced microhardness of the matrix, thus improving the bond strength of steel fibers.

*3.3. ASR Expansion Results*

Figure 5 shows the ASR induced expansion results of the tested UHPC mixtures. All results reported in Figure 5 show the average value obtained on five identical specimens with COV of less than 2%, which is within the ASTM C1260 limits. Mortar bar specimens incorporating CA (UHPC1) incurred 0.049% and 0.080% expansion at 14 and 28 days, respectively. Similarly, mortar bar specimens made with RS exhibited expansion of 0.045% and 0.070% at 14 and 28 days, respectively. The lower expansion of UHPC specimens can be attributed to its denser microstructure exhibiting negligible permeability, leading to restricted intrusion of external alkalis into the cementitious matrix. This can mitigate the formation of ASR gel and the associated damage propagation [48,69]. Further, the incorporation of CA and RS reduced the porosity by converting unstable calcium hydroxide (CH) into more stable secondary CSH [57,70,71]. It was observed that the incorporation of steel fibers in UHPC specimens resulted in remarkably lower ASR expansion compared to that of identical specimens without steel fibers. For instance, specimens incorporating CA and steel fibers (UHPFRC1) attained expansion of 0.0042% and 0.0071% at 14 and 28 days, respectively. Similarly, 0.0038% and 0.0065% expansion at 14 and 28 days, respectively, was observed for specimens made with RS and steel fibers (UHPFRC2). This corresponds to ten times lower expansion of UHPC specimens made with steel fibers than that of identical UHPC specimen without steel fiber.

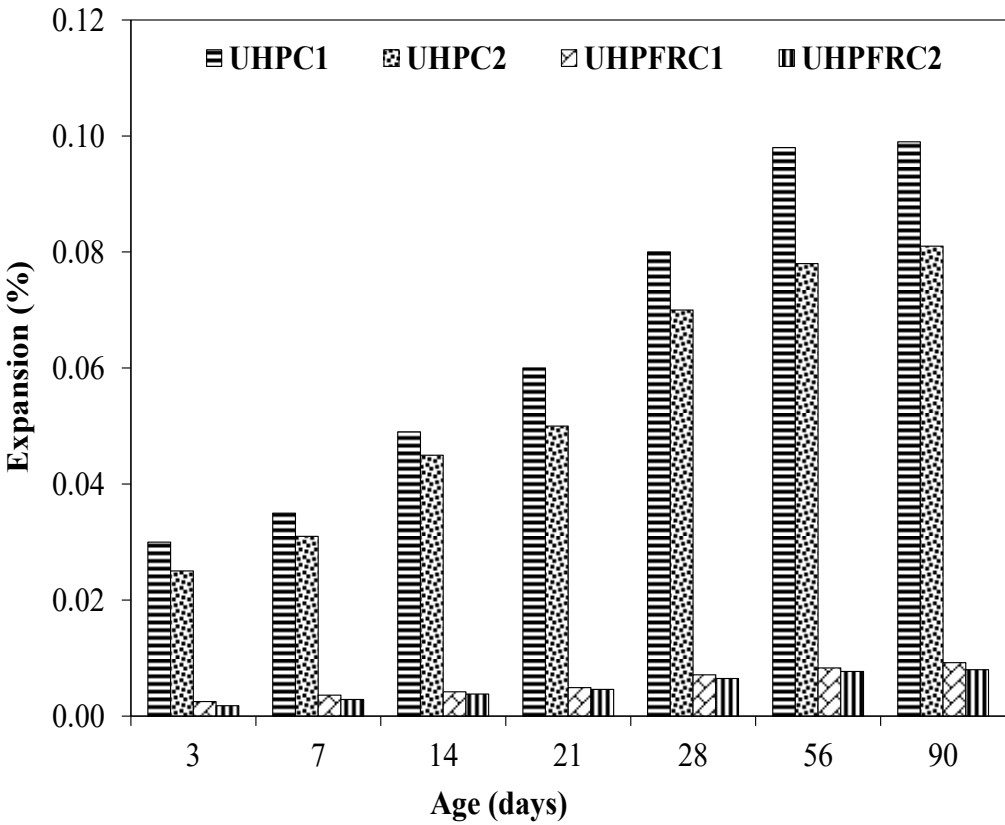

**Figure 5.** Expansion results for various UHPC mixtures.

Normally, steel fibers provide a restraint against the deterioration caused by the expansive ASR materials, which restrained the formation and propagation of microcracks. Moreover, the confinement provided by the fibers to the surrounding matrix restrains the distress caused by ASR gel, leading to localization of associated reactive spots and limiting the silica dissolution [72]. Moreover, the chemo-mechanical confinement effect of steel fibers further restricts the transportation of the developed aggressive ASR products to other locations and the change of the concentration of soluble ions [73]. Since, no significant evidence related to ASR associated damage has been observed in UHPC specimens owing to its denser and more compact microstructure, additional benefits of steel fibers inclusion leads to further enhanced material characteristics of UHPC against severe ASR exposure conditions. Furthermore, all the tested UHPC mixtures exhibited expansion of less than 0.01% at 90 days under ASR conducive exposure conditions. This indicates the superior long-term durability performance of the sustainable UHPC mixtures investigated.

According to ASTM C1260 [51], mortar bar specimens which exhibit expansion greater than 0.10% and 0.20% at 14 and 28 days, respectively, are considered as ASR reactive. For concrete mixture incorporating cement and other pozzolanic materials, expansion greater than 0.10% at 14 days are likely to be potentially reactive as per ASTM C1567 (2013) [74]. Similarly, according to RILEM AAR2 (2003) [75], accelerated mortar bar specimens showing expansion of less than 0.10% at 14 days of exposure are considered as non-reactive, while specimens exhibiting expansion greater than 0.20% are considered as reactive. According to AS 1141.60.1 (2014) [76], if the mortar bar expansion is greater than 0.10% and 0.30% at 10 and 21 days of exposure to NaOH solution at 80 °C, respectively, the aggregates can be considered as reactive. In the present study, all the tested UHPC specimens showed expansion of less than 0.014% and 0.030% at 28 days and 90 days, respectively. Therefore, it can be argued that UHPC showed no significant potential of ASR and associated damages. Similar findings were reported in previous study conducted on UHPC. For instance, An (2017) reported expansion of 0.0031% and 0.0041% at 14 and 28 days respectively, for an UHPC mixture incorporating rice husk ash [69].

### 3.4. Effect of ASR Exposure on Residual Compressive and Flexural Strength

Figure 6 depicts the effects of exposure to accelerated ASR conditions on the residual compressive and flexural strengths of the tested UHPC mixtures. It can be observed that the compressive strength of UHPC specimens subjected to accelerated ASR conditions increased, rather than the expected decrease expected for NSC. For example, the compressive strength of the UHPFRC2 specimens exposed to the water curing condition was 141 MPa in comparison to 150 MPa for identical specimens subjected to ASR conditions at 28 days. The results reveal that there was relatively higher increase in compressive strength at early age (i.e., 3 days) for UHPC mixtures subjected to accelerated ASR conditions as compared to that at later ages (i.e., 90 days). For instance, around 20% increase in compressive strength was observed at early age in comparison to 5% increase in compressive strength at later ages. This gain in compressive strength due to ASR exposure is mainly related to its exposure at higher temperature (80 °C) [56]. Higher temperature expedites the hydration and pozzolanic reactions process, leading to enhanced early gain in strength [77–79]. It should be noted that normally ASR exposure conditions have detrimental effect on the mechanical properties of concrete [42,43]. However, due to the near zero porosity of UHPC, the alkalis from the external source will be inhibited from entering inside the matrix, which mitigates their effect the internal microstructure. Similar results were observed for the flexural strength of tested UHPC mixtures, which improved upon accelerated ASR exposure. Therefore, the sustainable UHPC tested in this study is a promising alternative for construction in severe ASR conditions to achieve more durable and sustainable infrastructure.

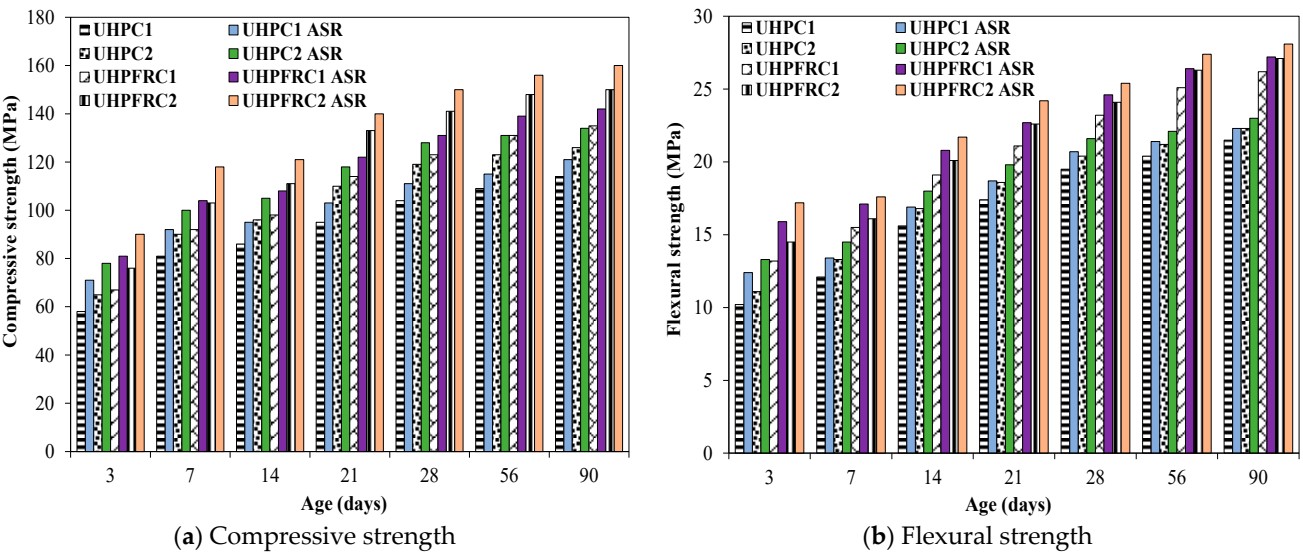

(**a**) Compressive strength       (**b**) Flexural strength

**Figure 6.** Effect of ASR conditions on compressive and flexural strength of UHPC mixtures.

### 3.5. Comparison of Ultra-High Perfromance Concrete with Normal Strength Concrete

Table 6 illustrates a comparison between the compressive and flexural strengths, and ASR expansion values for the tested UHPC mixtures and a benchmark NSC at different ages. The NSC specimens suffered expansion of 0.23, 0.28, and 0.36 at 14, 28, and 90 days, respectively, which is higher than the specified limits of ASTM C1260, indicating its vulnerability to ASR damage. On the other hand, UHPFRC2 specimens attained one order of magnitude lower expansion values of 0.0038, 0.0065, and 0.0080 at 14, 28, and 90 days, respectively. The expansion of UHPC specimens is also well below the specified limits by various standards [51,75,76]. It should be noted that UHPFRC2 showed negligible porosity in comparison with NSC, leading to decreased penetration of alkalis from the external NaOH solution to form ASR gel, thus inhibiting expansion and associate cracks [48].

**Table 6.** Comparison of mechanical strength and expansion results for UHPC and NSC.

| Properties | Exposure | UHPFRC2 (Days) | | | NSC (Days) | | |
|---|---|---|---|---|---|---|---|
| | | **14** | **28** | **90** | **14** | **28** | **90** |
| Expansion * (%) | ASR | 0.0038 | 0.0065 | 0.0080 | 0.230 | 0.280 | 0.361 |
| Compressive strength (MPa) | Water | 111 | 141 | 150 | 26.8 | 29.1 | 37.2 |
| | ASR [¶] | 121 | 150 | 160 | 24.5 | 26.5 | 33 |
| Flexural strength (MPa) | Water | 20 | 24 | 27 | 8.4 | 10.1 | 11.5 |
| | ASR [¶] | 21.7 | 25.4 | 28 | 8.0 | 9.5 | 10.0 |

* Expansion was measured for specimens subjected to accelerated ASR conducive conditions. [¶] ASR = Condition in which specimens were exposed to 1N NaOH solution at 80 °C.

The NSC specimens attained compressive strength of 26.8 MPa, 29.1 MPa, and 37.2 MPa at 14, 28, and 90 days, respectively. The respective values decreased to 24.5 MPa, 26.5 MPa, and 33 MPa when NSC specimens were subjected to accelerated ASR conditions, indicating the damaging effect of ASR expansive products on NSC. Furthermore, surface map cracks were visually observed on the tested NSC specimens (Figure 7a), confirming the formation of expansive ASR products. The ASR gel formed due to the reaction of alkalis and reactive silica causes localized swelling, leading to internal micro-cracking and decreasing the strength properties of concrete [43]. Further, Simen et al. [80] reported that the anisotropic stress caused by swelling due to ASR leads to crack formation and early failure of specimens. In another study, 10 micron sized cracks were observed, which propagated from the aggregate to matrix due to ASR [43]. The degradation of the compressive strength of NSC due to ASR have been well documented in previous studies [42,43]. On the other hand, interestingly, an increase in compressive strength of UHPFRC2 specimens exposed to ASR conditions was observed, demonstrating the insignificant effect of accelerated ASR exposure conditions. Further, no cracks could be observed at the surface of the tested UHPC specimens exposed to the accelerated ASR condition (Figure 7b).

The flexural strength of NSC specimens also decreased from 8.4, 10.1, and 11.5 MPa to 8.0, 9.5, and 10 MPa at 14, 28, and 90 days, respectively, upon exposure to accelerated ASR conditions. The reduction in flexural strength due to ASR exposure was primarily due to the additional tensile stresses exerted by the expansion of the ASR gel [42,43]. However, similar to compressive strength, an increase in flexural strength was observed for UHPC specimens exposed to accelerated ASR conditions. This increase in mechanical strength properties of UHPC subjected to accelerated ASR conditions can be related to the development of denser microstructure owing to the formation of more CSH products at elevated temperature [81]. SEM analysis showed the development of microcracks and ASR gel network in the tested NSC specimens subjected to ASR conditions (Figure 8). However, no microcracks or ASR gel could be retrieved in the UHPC specimens.

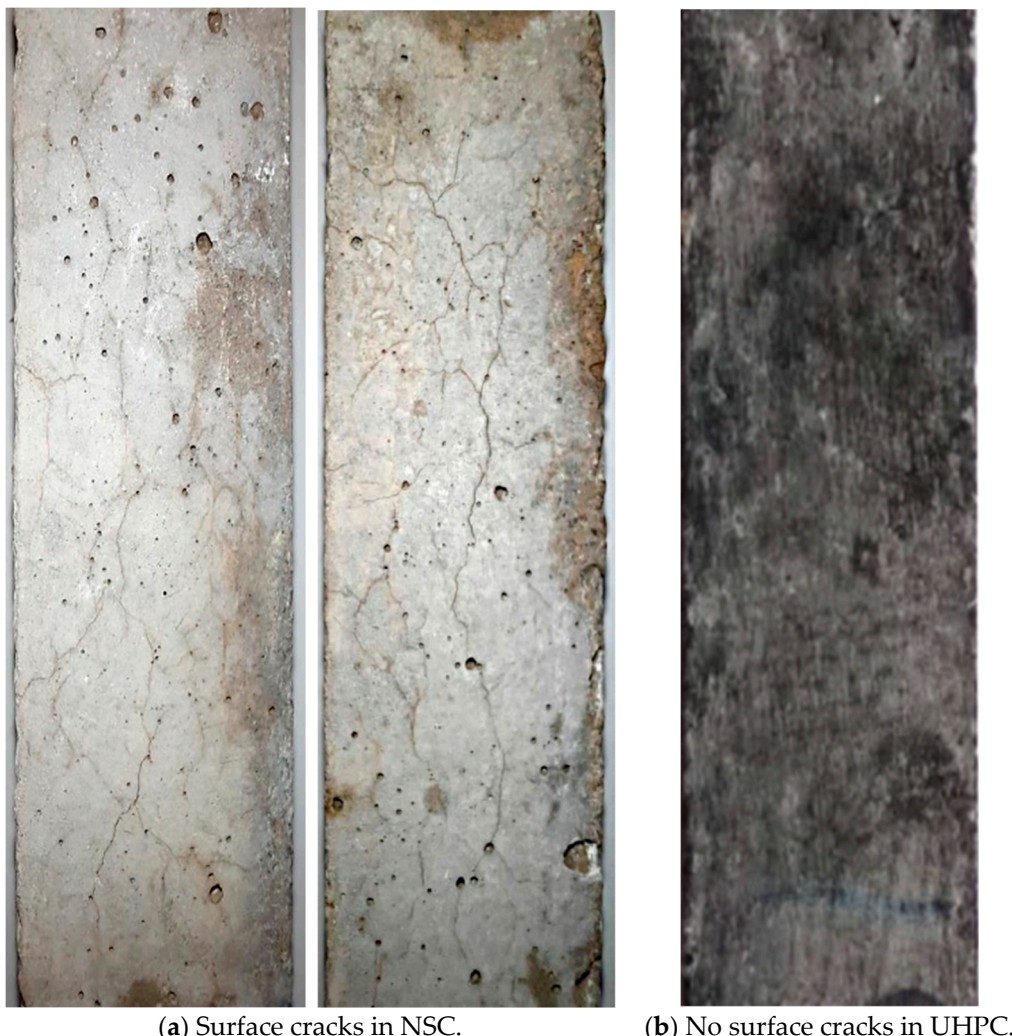

(**a**) Surface cracks in NSC.  (**b**) No surface cracks in UHPC.

**Figure 7.** Surface cracks in NSC and UHPC specimens exposed to accelerated ASR conductive conditions.

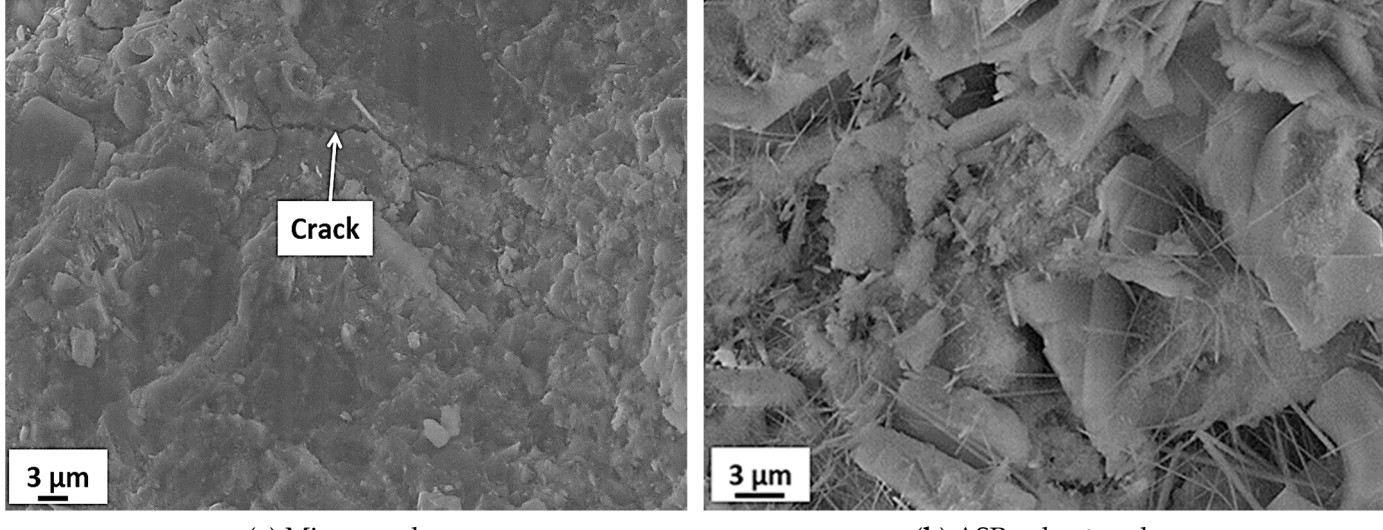

(**a**) Micro-crack.  (**b**) ASR gel network.

**Figure 8.** Micro-structural analysis of NSC.

## 4. Conclusions

The present study explored the mechanical and durability performance of sustainable UHPC mixtures incorporating CA and RS and made with locally recycled steel fibers under water and accelerated ASR conditions. The performance of UHPC mixtures was further compared with that of benchmark NSC. Based on the experimental results, the following conclusions can be drawn:

(1) Flowability of UHPC mixtures incorporating CA and RS was 186 and 195 mm, respectively. This decrease in flow of UHPC mixture with CA was mainly attributed to the porous structure of CA. Moreover, a decrease of around 10% in flow was observed for similar mixture due to incorporation of steel fibers.

(2) UHPC specimen with CA showed compressive strength of around 104 MPa at 28 days. UHPC specimen incorporating RS has higher compressive strength as compared to UHPC specimen with CA. For instance, compressive strength of UHPC specimen incorporating RS was around 14% higher than the UHPC mixture with CA at 28 days. Furthermore, compressive strength of UHPC specimens incorporating CA and RS improved with addition of locally produced steel fibers. It was observed that compressive strength of NSC specimen decreased under ASR conditions. However, interestingly, an increase in compressive strength of UHPC mixtures incorporating CA and RS exposed to accelerated ASR conditions was observed due to increased pozzolanic activity and negligible porosity of matrix with enhanced packing density at elevated temperature.

(3) The flexural strength of UHPC specimens made with CA was higher than 19 MPa at 28 days. It was observed that the flexural strength of UHPC specimens at 28 days increased by around 20% with the addition of recycled steel fibers owing to restraining microcracks via fiber bridging and strong fiber–matrix interaction, as confirmed by microstructural analysis.

(4) Results show that the flexural strength of UHPC specimens incorporating RS and CA and made with steel fibers under accelerated ASR conditions was 6% higher than that of identical specimens cured under normal water condition.

(5) The expansion of UHPC mortar bar specimens incorporating CA and RS under accelerated ASR conditions was well below the specified limits of various standards (ASTM C1260, RILEM AAR2, AS 1141.60.1). For example, all the tested UHPC specimens exhibited expansion of less than 0.014% at 28 days (which is lower than the specified limit of 0.20% by ASTM C1260, in comparison to the expansion of 0.28% for the benchmark NSC specimens.

(6) Microstructural analysis of NSC specimens subjected to accelerated ASR conditions showed micro-cracks due to ASR expansion. Conversely, no microcracking was detected for the tested UHPC. Rather, a strong fiber–matrix interaction was observed through SEM analysis under ASR exposure.

(7) The comparison between the performance of the UHPC and NSC specimens revealed that UHPC mixtures exposed to accelerated ASR conditions exhibited excellent characteristics owing to its improved microstructure and negligible porosity. While NSR incurred strength degradation and visual surface damage, sustainable UHPC specimens gained enhanced mechanical properties and exhibited no signs of damage.

(8) Therefore, sustainable UHPC mixtures incorporating recycled local waste materials along with locally produced steel fibers from recycled steel wire can be a strong contender as an eco-efficient and resilient construction material for sustainable and economical infrastructure developments. However, a detailed investigation of full-scale sustainable UHPC structural members under long exposure to aggressive environmental conditions is warranted in future investigations to give the local industry confidence in the advantages of this construction alternative.

(9) This study provides a benchmark, motivation and confidence to the construction industry for utilizing UHPC made from local recycled waste materials for use in full-scale structures, leading to durable and sustainable infrastructure development,

especially under aggressive ASR conditions. Furthermore, due to its remarkable mechanical and durability performance, UHPC requires low maintenance and repair expenses, leading to significant reduction in structural life cycle cost.

**Author Contributions:** Conceptualization, S.A., M.L.N. and W.A.; methodology, A.A., W.A., M.Y. and S.A.; analysis, S.A. and W.A.; investigation, A.A., M.Y. and W.A.; resources, S.A. and A.A.; writing—original draft preparation, S.A., W.A., A.A. and M.Y.; writing—review and editing, M.L.N. and S.A.; supervision, S.A. and M.L.N.; funding acquisition, S.A. and A.A. All authors have read and agreed to the published version of the manuscript.

**Funding:** This research was a part of the HEC-NRPU 9820 research project.

**Data Availability Statement:** The data presented in this study are available on request from the corresponding author.

**Acknowledgments:** The authors acknowledge the support and facilities of the Civil Engineering Laboratories at the University of Engineering and Technology, Lahore.

**Conflicts of Interest:** The authors declare no conflict of interest.

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
