# Peer review of "Investigation of Alkali-Silica Reactivity in Sustainable Ultrahigh Performance Concrete"

_sustainability, doi:10.3390/su13105680_

Round 1

Reviewer 1 Report

Congratulations to the authors of a very interesting and up-to-date study on the assessment of concrete resistance to alkaline aggression.

Very important conclusion is that comparison between the performance of the UHPC and NSC specimens revealed that UHPC mixtures exposed to accelerated ASR conditions exhibited excellent characteristics owing to its improved microstructure and negligible porosity.

While NSR incurred strength degradation and visual surface damage, sustainable UHPC specimens gained enhanced mechanical properties and exhibited no signs of damage.

The reviewed article has great scientific value and the conclusions can be implemented in the industry.

Author Response

The authors appreciate the reviewer’s encouraging and positive feedback on the conducted study. They command the reviewer for identifying the important findings of the study. The authors are grateful to the reviewer for the very encouraging feedback.

Reviewer 2 Report

Comments

This paper studied alkali-silica reactivity in sustainable ultrahigh performance concrete. The outcome is interesting for readers. However, there are several aspects that need to be improved. The reviewer can only recommend for publication if the author satisfactorily address the following major comments in the revised version.

  1. Why the amount of silica was not varied in Table 3 while the study was focused on ASR?
  2. Suggest to provide standard deviation of the results in Table 1.
  3. How compressive and flexural strength is connected with ASR? This need to be explained.
  4. The failure mechanism of the specimen should be discussed more clearly.
  5. The novelty of the study should be highlighted more clearly at the end of introduction section. How this study is different from the published study in literature?
  6. How the outcome of this study will benefit researchers and end users? This need to be highlighted in introduction or end of conclusion.
  7. The recent progress on the durability of concrete should be highlighted in introduction section to improve the background study. The recent investigation of concrete properties under microscope [Ref: Characteristics, strength development and microstructure of cement mortar containing oil-contaminated sand], hygrothermal [Ref: Ageing of particulate-filled epoxy resin under hygrothermal conditions] and UV [Ref: Effects of ultraviolet solar radiation on the properties of particulate-filled epoxy based polymer coating] conditions indicated the durability concern of concrete material. Suggest to include them in introduction section with proper citations to improve the background study.

I would be happy to see the revised version to understand how these comments are being addressed.

Author Response

The authors thank the reviewer for the insightful feedback and constructive comments, which enhanced the quality of the revised manuscript. We have carefully addressed each comment of the reviewer. A detailed explanation of how the authors have addressed each comment of the reviewer is attached.

Round 2

Reviewer 2 Report

I have no further comments